# LEARNING END-TO-END GOAL-ORIENTED DIALOG

**Antoine Bordes, Y-Lan Boureau & Jason Weston**
Facebook AI Research
New York, USA
`{abordes, ylan, jase}@fb.com`

## ABSTRACT

Traditional dialog systems used in goal-oriented applications require a lot of domain-specific handcrafting, which hinders scaling up to new domains. End-to-end dialog systems, in which all components are trained from the dialogs themselves, escape this limitation. But the encouraging success recently obtained in chit-chat dialog may not carry over to goal-oriented settings. This paper proposes a testbed to break down the strengths and shortcomings of end-to-end dialog systems in goal-oriented applications. Set in the context of restaurant reservation, our tasks require manipulating sentences and symbols in order to properly conduct conversations, issue API calls and use the outputs of such calls. We show that an end-to-end dialog system based on Memory Networks can reach promising, yet imperfect, performance and learn to perform non-trivial operations. We confirm those results by comparing our system to a hand-crafted slot-filling baseline on data from the second Dialog State Tracking Challenge (Henderson *et al.*, 2014a). We show similar result patterns on data extracted from an online concierge service.

## 1  INTRODUCTION

The most useful applications of dialog systems such as digital personal assistants or bots are currently goal-oriented and transactional: the system needs to understand a user request and complete a related task with a clear goal within a limited number of dialog turns. The workhorse of traditional dialog systems is *slot-filling* (Lemon *et al.*, 2006; Wang and Lemon, 2013; Young *et al.*, 2013) which predefines the structure of a dialog state as a set of slots to be filled during the dialog. For a restaurant reservation system, such slots can be the location, price range or type of cuisine of a restaurant. Slot-filling has proven reliable but is inherently hard to scale to new domains: it is impossible to manually encode all features and slots that users might refer to in a conversation.

End-to-end dialog systems, usually based on neural networks (Shang *et al.*, 2015; Vinyals and Le, 2015; Sordoni *et al.*, 2015; Serban *et al.*, 2015a; Dodge *et al.*, 2016), escape such limitations: all their components are directly trained on past dialogs, with no assumption on the domain or dialog state structure, thus making it easy to automatically scale up to new domains. They have shown promising performance in non goal-oriented *chit-chat* settings, where they were trained to predict the next utterance in social media and forum threads (Ritter *et al.*, 2011; Wang *et al.*, 2013; Lowe *et al.*, 2015) or movie conversations (Banchs, 2012). But the performance achieved on chit-chat may not necessarily carry over to goal-oriented conversations. As illustrated in Figure 1 in a restaurant reservation scenario, conducting goal-oriented dialog requires skills that go beyond language modeling, e.g., asking questions to clearly define a user request, querying Knowledge Bases (KBs), interpreting results from queries to display options to users or completing a transaction. This makes it hard to ascertain how well end-to-end dialog models would do, especially since evaluating chit-chat performance in itself is not straightforward (Liu *et al.*, 2016). In particular, it is unclear if end-to-end models are in a position to replace traditional dialog methods in a goal-directed setting: can end-to-end dialog models be competitive with traditional methods even in the well-defined narrow-domain tasks where they excel? If not, where do they fall short?

This paper aims to make it easier to address these questions by proposing an open resource to test end-to-end dialog systems in a way that 1) favors reproducibility and comparisons, and 2) is lightweight and easy to use. We aim to break down a goal-directed objective into several subtasks to test some crucial capabilities that dialog systems should have (and hence provide error analysis by design).

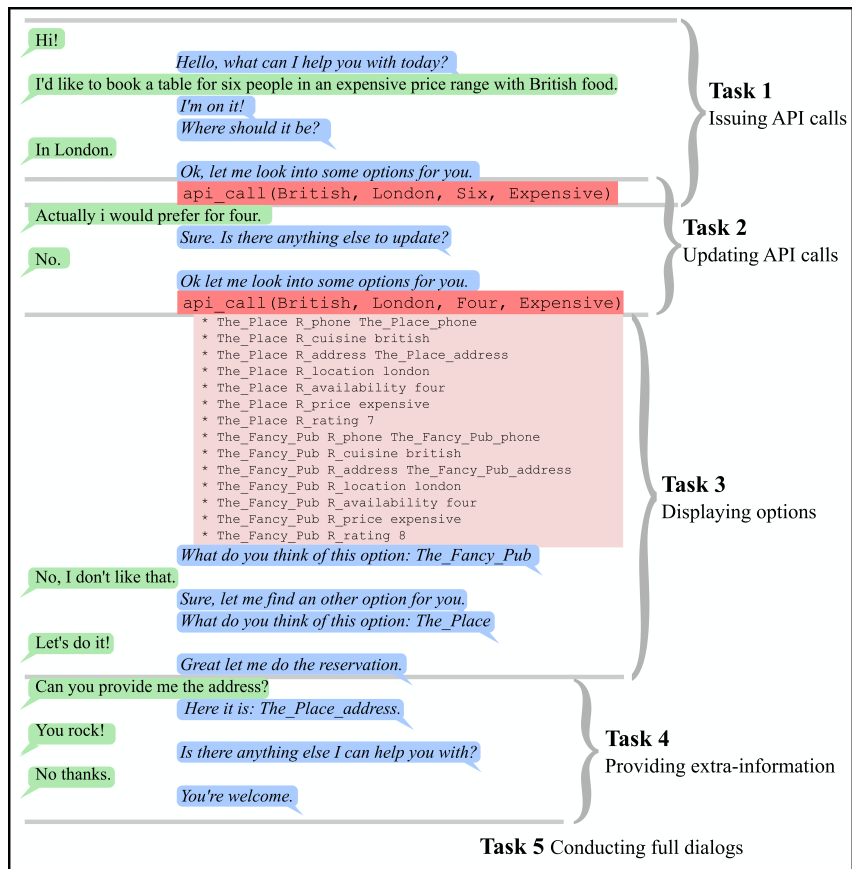

Figure 1: **Goal-oriented dialog tasks.** A user (in green) chats with a bot (in blue) to book a table at a restaurant. Models must predict bot utterances and API calls (in dark red). Task 1 tests the capacity of interpreting a request and asking the right questions to issue an API call. Task 2 checks the ability to modify an API call. Task 3 and 4 test the capacity of using outputs from an API call (in light red) to propose options (sorted by rating) and to provide extra-information. Task 5 combines everything.

In the spirit of the bAbI tasks conceived as question answering testbeds (Weston *et al.*, 2015b), we designed a set of five tasks within the goal-oriented context of restaurant reservation. Grounded with an underlying KB of restaurants and their properties (location, type of cuisine, etc.), these tasks cover several dialog stages and test if models can learn various abilities such as performing dialog management, querying KBs, interpreting the output of such queries to continue the conversation or dealing with new entities not appearing in dialogs from the training set. In addition to showing how the set of tasks we propose can be used to test the goal-directed capabilities of an end-to-end dialog system, we also propose results on two additional datasets extracted from real interactions with users, to confirm that the pattern of results observed in our tasks is indeed a good proxy for what would be observed on real data, with the added benefit of better reproducibility and interpretability.

The goal here is explicitly not to improve the state of the art in the narrow domain of restaurant booking, but to take a narrow domain where traditional handcrafted dialog systems are known to perform well, and use that to gauge the strengths and weaknesses of current end-to-end systems with no domain knowledge. Solving our tasks requires manipulating both natural language and symbols from a KB. Evaluation uses two metrics, per-response and per-dialog accuracies, the latter tracking completion of the actual goal. Figure 1 depicts the tasks and Section 3 details them. Section 4 compares multiple methods on these tasks. As an end-to-end neural model, we tested Memory Networks (Weston *et al.*, 2015a), an attention-based architecture that has proven competitive for non goal-oriented dialog (Dodge *et al.*, 2016). Our experiments in Section 5 show that Memory Networks can be trained to perform non-trivial operations such as issuing API calls to KBs and manipulating entities unseen in training. We confirm our findings on real human-machine dialogs

Table 1: **Data used in this paper.** Tasks 1-5 were generated using our simulator and share the same KB. Task 6 was converted from the $2^{nd}$ Dialog State Tracking Challenge (Henderson *et al.*, 2014a). *Concierge* is made of chats extracted from a real online concierge service. [*] Tasks 1-5 have two test sets, one using the vocabulary of the training set and the other using out-of-vocabulary words.

| | Tasks | T1 | T2 | T3 | T4 | T5 | T6 | Concierge |
|---|---|---|---|---|---|---|---|---|
| | Number of utterances: | 12 | 17 | 43 | 15 | 55 | 54 | 8 |
| DIALOGS | - user utterances | 5 | 7 | 7 | 4 | 13 | 6 | 4 |
| *Average statistics* | - bot utterances | 7 | 10 | 10 | 4 | 18 | 8 | 4 |
| | - outputs from API calls | 0 | 0 | 23 | 7 | 24 | 40 | 0 |
| | Vocabulary size | | | 3,747 | | | 1,229 | 8,629 |
| | Candidate set size | | | 4,212 | | | 2,406 | 11,482 |
| DATASETS | Training dialogs | | | 1,000 | | | 1,618 | 3,249 |
| *Tasks 1-5 share the* | Validation dialogs | | | 1,000 | | | 500 | 403 |
| *same data source* | Test dialogs | | | 1,000[*] | | | 1,117 | 402 |

from the restaurant reservation dataset of the $2^{nd}$ Dialog State Tracking Challenge, or DSTC2 (Henderson *et al.*, 2014a), which we converted into our task format, showing that Memory Networks can outperform a dedicated slot-filling rule-based baseline. We also evaluate on a dataset of human-human dialogs extracted from an online concierge service that books restaurants for users. Overall, the per-response performance is encouraging, but the per-dialog one remains low, indicating that end-to-end models still need to improve before being able to reliably handle goal-oriented dialog.

## 2 RELATED WORK

The most successful goal-oriented dialog systems model conversation as partially observable Markov decision processes (POMDP) (Young *et al.*, 2013). However, despite recent efforts to learn modules (Henderson *et al.*, 2014b), they still require many hand-crafted features for the state and action space representations, which restrict their usage to narrow domains. Our simulation, used to generate goal-oriented datasets, can be seen as an equivalent of the user simulators used to train POMDP (Young *et al.*, 2013; Pietquin and Hastie, 2013), but for training end-to-end systems.

Serban *et al.* (2015b) list available corpora for training dialog systems. Unfortunately, no good resources exist to train and test end-to-end models in goal-oriented scenarios. Goal-oriented datasets are usually designed to train or test dialog state tracker components (Henderson *et al.*, 2014a) and are hence of limited scale and not suitable for end-to-end learning (annotated at the state level and noisy). However, we do convert the Dialog State Tracking Challenge data into our framework. Some datasets are not open source, and require a particular license agreement or the participation to a challenge (e.g., the end-to-end task of DSTC4 (Kim *et al.*, 2016)) or are proprietary (e.g., Chen *et al.* (2016)). Datasets are often based on interactions between users and existing systems (or ensemble of systems) like DSTC datasets, SFCore (Gašic *et al.*, 2014) or ATIS (Dahl *et al.*, 1994). This creates noise and makes it harder to interpret the errors of a model. Lastly, resources designed to connect dialog systems to users, in particular in the context of reinforcement learning, are usually built around a crowdsourcing setting such as Amazon Mechanical Turk, e.g., (Hixon *et al.*, 2015; Wen *et al.*, 2015; Su *et al.*, 2015a;b). While this has clear advantages, it prevents reproducibility and consistent comparisons of methods in the exact same setting.

The closest resource to ours might be the set of tasks described in (Dodge *et al.*, 2016), since some of them can be seen as goal-oriented. However, those are question answering tasks rather than dialog, i.e. the bot only responds with answers, never questions, which does not reflect full conversation.

## 3 GOAL-ORIENTED DIALOG TASKS

All our tasks involve a restaurant reservation system, where the goal is to book a table at a restaurant. The first five tasks are generated by a simulation, the last one uses real human-bot dialogs. The data for all tasks is available at `http://fb.ai/babi`. We also give results on a proprietary dataset extracted from an online restaurant reservation concierge service with anonymized users.

## 3.1 RESTAURANT RESERVATION SIMULATION

The simulation is based on an underlying KB, whose facts contain the restaurants that can be booked and their properties. Each restaurant is defined by a type of cuisine (10 choices, e.g., French, Thai), a location (10 choices, e.g., London, Tokyo), a price range (cheap, moderate or expensive) and a rating (from 1 to 8). For simplicity, we assume that each restaurant only has availability for a single party size (2, 4, 6 or 8 people). Each restaurant also has an address and a phone number listed in the KB.

The KB can be queried using API calls, which return the list of facts related to the corresponding restaurants. Each query must contain four fields: a location, a type of cuisine, a price range and a party size. It can return facts concerning one, several or no restaurant (depending on the party size).

Using the KB, conversations are generated in the format shown in Figure 1. Each example is a dialog comprising utterances from a user and a bot, as well as API calls and the resulting facts. Dialogs are generated after creating a user request by sampling an entry for each of the four required fields: e.g. the request in Figure 1 is [cuisine: British, location: London, party size: six, price range: expensive]. We use natural language patterns to create user and bot utterances. There are 43 patterns for the user and 20 for the bot (the user can use up to 4 ways to say something, while the bot always uses the same). Those patterns are combined with the KB entities to form thousands of different utterances.

### 3.1.1 TASK DEFINITIONS

We now detail each task. Tasks 1 and 2 test dialog management to see if end-to-end systems can learn to implicitly track dialog state (never given explicitly), whereas Task 3 and 4 check if they can learn to use KB facts in a dialog setting. Task 3 also requires to learn to sort. Task 5 combines all tasks.

**Task 1: Issuing API calls**    A user request implicitly defines a query that can contain from 0 to 4 of the required fields (sampled uniformly; in Figure 1, it contains 3). The bot must ask questions for filling the missing fields and eventually generate the correct corresponding API call. The bot asks for information in a deterministic order, making prediction possible.

**Task 2: Updating API calls**    Starting by issuing an API call as in Task 1, users then ask to update their requests between 1 and 4 times (sampled uniformly). The order in which fields are updated is random. The bot must ask users if they are done with their updates and issue the updated API call.

**Task 3: Displaying options**    Given a user request, we query the KB using the corresponding API call and add the facts resulting from the call to the dialog history. The bot must propose options to users by listing the restaurant names sorted by their corresponding rating (from higher to lower) until users accept. For each option, users have a 25% chance of accepting. If they do, the bot must stop displaying options, otherwise propose the next one. Users always accept the option if this is the last remaining one. We only keep examples with API calls retrieving at least 3 options.

**Task 4: Providing extra information**    Given a user request, we sample a restaurant and start the dialog as if users had agreed to book a table there. We add all KB facts corresponding to it to the dialog. Users then ask for the phone number of the restaurant, its address or both, with proportions 25%, 25% and 50% respectively. The bot must learn to use the KB facts correctly to answer.

**Task 5: Conducting full dialogs**    We combine Tasks 1-4 to generate full dialogs just as in Figure 1. Unlike in Task 3, we keep examples if API calls return at least 1 option instead of 3.

### 3.1.2 DATASETS

We want to test how well models handle entities appearing in the KB but not in the dialog training sets. We split types of cuisine and locations in half, and create two KBs, one with all facts about restaurants within the first halves and one with the rest. This yields two KBs of 4,200 facts and 600 restaurants each (5 types of cuisine $\times$ 5 locations $\times$ 3 price ranges $\times$ 8 ratings) that only share price ranges, ratings and party sizes, but have disjoint sets of restaurants, locations, types of cuisine, phones and addresses. We use one of the KBs to generate the standard training, validation and test dialogs, and use the other KB only to generate test dialogs, termed Out-Of-Vocabulary (OOV) test sets.

For training, systems have access to the training examples and both KBs. We then evaluate on both test sets, plain and OOV. Beyond the intrinsic difficulty of each task, the challenge on the OOV test

sets is for models to generalize to new entities (restaurants, locations and cuisine types) unseen in any training dialog – something natively impossible for embedding methods. Ideally, models could, for instance, leverage information coming from the entities of the same type seen during training.

We generate five datasets, one per task defined in 3.1.1. Table 1 gives their statistics. Training sets are relatively small (1,000 examples) to create realistic learning conditions. The dialogs from the training and test sets are different, never being based on the same user requests. Thus, we test if models can generalize to new combinations of fields. Dialog systems are evaluated in a ranking, not a generation, setting: at each turn of the dialog, we test whether they can predict bot utterances and API calls by selecting a candidate, not by generating it.[1] Candidates are ranked from a set of all bot utterances and API calls appearing in training, validation and test sets (plain and OOV) for all tasks combined.

## 3.2   DIALOG STATE TRACKING CHALLENGE

Since our tasks rely on synthetically generated language for the user, we supplement our dataset with real human-bot dialogs. We use data from DSTC2 (Henderson *et al.*, 2014a), that is also in the restaurant booking domain. Unlike our tasks, its user requests only require 3 fields: type of cuisine (91 choices), location (5 choices) and price range (3 choices). The dataset was originally designed for dialog state tracking hence every dialog turn is labeled with a state (a user intent + slots) to be predicted. As our goal is to evaluate end-to-end training, we did not use that, but instead converted the data into the format of our 5 tasks and included it in the dataset as Task 6.

We used the provided speech transcriptions to create the user and bot utterances, and given the dialog states we created the API calls to the KB and their outputs which we added to the dialogs. We also added ratings to the restaurants returned by the API calls, so that the options proposed by the bots can be consistently predicted (by using the highest rating). We did use the original test set but use a slightly different training/validation split. Our evaluation differs from the challenge (we do not predict the dialog state), so we cannot compare with the results from (Henderson *et al.*, 2014a).

This dataset has similar statistics to our Task 5 (see Table 1) but is harder. The dialogs are noisier and the bots made mistakes due to speech recognition errors or misinterpretations and also do not always have a deterministic behavior (the order in which they can ask for information varies).

## 3.3   ONLINE CONCIERGE SERVICE

Tasks 1-6 are, at least partially, artificial. This provides perfect control over their design (at least for Tasks 1-5), but no guarantee that good performance would carry over from such synthetic to more realistic conditions. To quantify this, we also evaluate the models from Section 4 on data extracted from a real online concierge service performing restaurant booking: users make requests through a text-based chat interface that are handled by human operators who can make API calls. All conversations are between native English speakers.

We collected around 4k chats to create this extra dataset, denoted *Concierge*. All conversations have been anonymized by (1) removing all user identifiers, (2) using the Stanford NER tagger to remove named entities (locations, timestamps, etc.), (3) running some manually defined regex to filter out any remaining salient information (phone numbers, etc.). The dataset does not contain results from API calls, but still records when operators made use of an external service (Yelp or OpenTable) to gather information. Hence, these have to be predicted, but without any argument (unlike in Task 2).

The statistics of *Concierge* are given in Table 1. The dialogs are shorter than in Tasks 1-6, especially since they do not include results of API calls, but the vocabulary is more diverse and so is the candidate set; the candidate set is made of all utterances of the operator appearing in the training, validation and test sets. Beyond the higher variability of the language used by human operators compared to bots, the dataset offers additional challenges. The set of user requests is much wider, ranging from managing restaurant reservations to asking for recommendations or specific information. Users do not always stay focused on the request. API calls are not always used (e.g., the operator might use neither Yelp nor OpenTable to find a restaurant), and facts about restaurants are not structured nor constrained as in a KB. The structure of dialogs is thus much more variable. Users and operators also make typos, spelling and grammar mistakes.

---

[1] Lowe *et al.* (2016) termed this setting Next-Utterance-Classification.

## 4 MODELS

To demonstrate how to use the dataset and provide baselines, we evaluate several learning methods on our goal-oriented dialog tasks: rule-based systems, classical information retrieval methods, supervised embeddings, and end-to-end Memory networks.

### 4.1 RULE-BASED SYSTEMS

Our tasks T1-T5 are built with a simulator so as to be completely predictable. Thus it is possible to hand-code a rule based system that achieves 100% on them, similar to the bAbI tasks of Weston *et al.* (2015b). Indeed, the point of these tasks is not to check whether a human is smart enough to be able to build a rule-based system to solve them, but to help analyze in which circumstances machine learning algorithms are smart enough to work, and where they fail.

However, the Dialog State Tracking Challenge task (T6) contains some real interactions with users. This makes rule-based systems less straightforward and not so accurate (which is where we expect machine learning to be useful). We implemented a rule-based system for this task in the following way. We initialized a dialog state using the 3 relevant slots for this task: cuisine type, location and price range. Then we analyzed the training data and wrote a series of rules that fire for triggers like word matches, positions in the dialog, entity detections or dialog state, to output particular responses, API calls and/or update a dialog state. Responses are created by combining patterns extracted from the training set with entities detected in the previous turns or stored in the dialog state. Overall we built 28 rules and extracted 21 patterns. We optimized the choice of rules and their application priority (when needed) using the validation set, reaching a validation per-response accuracy of 40.7%. We did not build a rule-based system for *Concierge* data as it is even less constrained.

### 4.2 CLASSICAL INFORMATION RETRIEVAL MODELS

Classical information retrieval (IR) models with no machine learning are standard baselines that often perform surprisingly well on dialog tasks (Isbell *et al.*, 2000; Jafarpour *et al.*, 2010; Ritter *et al.*, 2011; Sordoni *et al.*, 2015). We tried two standard variants:

**TF-IDF Match** For each possible candidate response, we compute a matching score between the input and the response, and rank the responses by score. The score is the TF–IDF weighted cosine similarity between the bag-of-words of the input and bag-of-words of the candidate response. We consider the case of the input being either only the last utterance or the entire conversation history, and choose the variant that works best on the validation set (typically the latter).

**Nearest Neighbor** Using the input, we find the most similar conversation in the training set, and output the response from that example. In this case we consider the input to only be the last utterance, and consider the training set as (utterance, response) pairs that we select from. We use word overlap as the scoring method. When several responses are associated with the same utterance in training, we sort them by decreasing co-occurence frequency.

### 4.3 SUPERVISED EMBEDDING MODELS

A standard, often strong, baseline is to use supervised word embedding models for scoring (conversation history, response) pairs. The embedding vectors are trained directly for this goal. In contrast, word embeddings are most well-known in the context of unsupervised training on raw text as in *word2vec* (Mikolov *et al.*, 2013). Such models are trained by learning to predict the middle word given the surrounding window of words, or vice-versa. However, given training data consisting of dialogs, a much more direct and strongly performing training procedure can be used: predict the next response given the previous conversation. In this setting a candidate reponse $y$ is scored against the input $x$: $f(x, y) = (Ax)^\top By$, where $A$ and $B$ are $d \times V$ word embedding matrices, i.e. input and response are treated as summed bags-of-embeddings. We also consider the case of enforcing $A = B$, which sometimes works better, and optimize the choice on the validation set.

The embeddings are trained with a margin ranking loss: $f(x, y) > m + f(x, \bar{y})$, with $m$ the size of the margin, and we sample $N$ *negative* candidate responses $\bar{y}$ per example, and train with SGD. This approach has been previously shown to be very effective in a range of contexts (Bai *et al.*, 2009;

Dodge *et al.*, 2016). This method can be thought of as a classical information retrieval model, but where the matching function is learnt.

## 4.4 MEMORY NETWORKS

Memory Networks (Weston *et al.*, 2015a; Sukhbaatar *et al.*, 2015) are a recent class of models that have been applied to a range of natural language processing tasks, including question answering (Weston *et al.*, 2015b), language modeling (Sukhbaatar *et al.*, 2015), and non-goal-oriented dialog (Dodge *et al.*, 2016). By first writing and then iteratively reading from a memory component (using *hops*) that can store historical dialogs and short-term context to reason about the required response, they have been shown to perform well on those tasks and to outperform some other end-to-end architectures based on Recurrent Neural Networks. Hence, we chose them as end-to-end model baseline.

We use the MemN2N architecture of Sukhbaatar *et al.* (2015), with an additional modification to leverage exact matches and types, described shortly. Apart from that addition, the main components of the model are (i) how it stores the conversation in memory, (ii) how it reads from the memory to reason about the response; and (iii) how it outputs the response. The details are given in Appendix A.

## 4.5 MATCH TYPE FEATURES TO DEAL WITH ENTITIES

Words denoting entities have two important traits: 1) exact matches are usually more appropriate to deal with them than approximate matches, and 2) they frequently appear as OOV words (e.g., the name of a new restaurant). Both are a challenge for embedding-based methods. Firstly, embedding into a low dimensional space makes it hard to differentiate between *exact* word matches, and matches between words with similar meaning (Bai *et al.*, 2009). While this can be a virtue (e.g. when using synonyms), it is often a flaw when dealing with entities (e.g. failure to differentiate between phone numbers since they have similar embeddings). Secondly, when a new word is used (e.g. the name of a new restaurant) not seen before in training, no word embedding is available, typically resulting in failure (Weston *et al.*, 2015a).

Both problems can be alleviated with match type features. Specifically, we augment the vocabulary with 7 special words, one for each of the KB entity types (cuisine type, location, price range, party size, rating, phone number and address). For each type, the corresponding type word is added to the candidate representation if a word is found that appears 1) as a KB entity of that type, 2) in the candidate, and 3) in the input or memory. Any word that matches as a KB entity can be typed even if it has never been seen before in training dialogs. These features allow the model to learn to rely on type information using exact matching words cues when OOV entity embeddings are not known, as long as it has access to a KB with the OOV entities. We assess the impact of such features for TF-IDF Match, Supervised Embeddings and Memory Networks.

## 5 EXPERIMENTS

Our main results across all the models and tasks are given in Table 2 (extra results are also given in Table 10 of Appendix D). The first 5 rows show tasks T1-T5, and rows 6-10 show the same tasks in the out-of-vocabulary setting. Rows 11 and 12 give results for the Dialog State Tracking Challenge task (T6) and Concierge respectively. Columns 2-7 give the results of each method tried in terms of per-response accuracy and per-dialog accuracy, the latter given in parenthesis. Per-response accuracy counts the percentage of responses that are correct (i.e., the correct candidate is chosen out of all possible candidates). Per-dialog accuracy counts the percentage of dialogs where every response is correct. Ultimately, if only one response is incorrect this could result in a failed dialog, i.e. failure to achieve the goal (in this case, of achieving a restaurant booking). Note that we test Memory Networks (MemNNs) with and without match type features, the results are shown in the last two columns. The hyperparameters for all models were optimized on the validation sets; values for best performing models are given in Appendix C.

The classical IR method TF-IDF Match performs the worst of all methods, and much worse than the Nearest Neighbor IR method, which is true on both the simulated tasks T1-T5 and on the real data of T6 and Concierge. Supplementing TF-IDF Match with match type features noticeably improves performance, which however still remains far behind Nearest Neighbor IR (adding bigrams to the

Table 2: **Test results across all tasks and methods.** For tasks T1-T5 results are given in the standard setup and the out-of-vocabulary (OOV) setup, where words (e.g. restaurant names) may not have been seen during training. Task T6 is the Dialog state tracking 2 task with real dialogs, and only has one setup. Best performing methods (or methods within 0.1% of best performing) are given in bold for the per-response accuracy metric, with the per-dialog accuracy given in parenthesis. [*] For Concierge, an example is considered correctly answered if the correct response is ranked among the top 10 candidates by the bot, to accommodate the much larger range of semantically equivalent responses among candidates (see ex. in Tab. 7) . [†] We did not implement MemNNs+match type on Concierge, because this method requires a KB and there is none associated with it.

| Task | Rule-based Systems | TF-IDF Match | | Nearest Neighbor | Supervised Embeddings | Memory Networks | |
| --- | --- | --- | --- | --- | --- | --- | --- |
| | | no type | + type | | | no match type | + match type |
| T1: Issuing API calls | *100 (100)* | 5.6 (0) | 22.4 (0) | 55.1 (0) | **100** (100) | **99.9** (99.6) | **100** (100) |
| T2: Updating API calls | *100 (100)* | 3.4 (0) | 16.4 (0) | 68.3 (0) | 68.4 (0) | **100** (100) | 98.3 (83.9) |
| T3: Displaying options | *100 (100)* | 8.0 (0) | 8.0 (0) | 58.8 (0) | 64.9 (0) | **74.9** (2.0) | **74.9** (0) |
| T4: Providing information | *100 (100)* | 9.5 (0) | 17.8 (0) | 28.6 (0) | 57.2 (0) | 59.5 (3.0) | **100** (100) |
| T5: Full dialogs | *100 (100)* | 4.6 (0) | 8.1 (0) | 57.1 (0) | 75.4 (0) | **96.1** (49.4) | 93.4 (19.7) |
| T1(OOV): Issuing API calls | *100 (100)* | 5.8 (0) | 22.4 (0) | 44.1 (0) | 60.0 (0) | 72.3 (0) | **96.5** (82.7) |
| T2(OOV): Updating API calls | *100 (100)* | 3.5 (0) | 16.8 (0) | 68.3 (0) | 68.3 (0) | 78.9 (0) | **94.5** (48.4) |
| T3(OOV): Displaying options | *100 (100)* | 8.3 (0) | 8.3 (0) | 58.8 (0) | 65.0 (0) | 74.4 (0) | **75.2** (0) |
| T4(OOV): Providing inform. | *100 (100)* | 9.8 (0) | 17.2 (0) | 28.6 (0) | 57.0 (0) | 57.6 (0) | **100** (100) |
| T5(OOV): Full dialogs | *100 (100)* | 4.6 (0) | 9.0 (0) | 48.4 (0) | 58.2 (0) | 65.5 (0) | **77.7** (0) |
| T6: Dialog state tracking 2 | 33.3 (0) | 1.6 (0) | 1.6 (0) | 21.9 (0) | 22.6 (0) | **41.1** (0) | **41.0** (0) |
| Concierge[*] | n/a | 1.1 (0.2) | n/a | 13.4 (0.5) | 14.6 (0.5) | **16.7** (1.2) | n/a[†] |

dictionary has no effect on performance). This is in sharp contrast to other recent results on data-driven *non*-goal directed conversations, e.g. over dialogs on Twitter (Ritter *et al.*, 2011) or Reddit (Dodge *et al.*, 2016), where it was found that TF-IDF Match outperforms Nearest Neighbor, as general conversations on a given subject typically share many words. We conjecture that the goal-oriented nature of the conversation means that the conversation moves forward more quickly, sharing fewer words per (input, response) pair, e.g. consider the example in Figure 1.

Supervised embeddings outperform classical IR methods in general, indicating that learning mappings between words (via word embeddings) is important. However, only one task (T1, Issuing API calls) is completely successful. In the other tasks, some responses are correct, as shown by the per-response accuracy, however there is no dialog where the goal is actually achieved (i.e., the mean dialog-accuracy is 0). Typically the model can provide correct responses for greeting messages, asking to wait, making API calls and asking if there are any other options necessary. However, it fails to interpret the results of API calls to display options, provide information or update the calls with new information, resulting in most of its errors, even when match type features are provided.

Memory Networks (without match type features) outperform classical IR and supervised embeddings across all of the tasks. They can solve the first two tasks (issuing and updating API calls) adequately. On the other tasks, they give improved results, but do not solve them. While the per-response accuracy is improved, the per-dialog accuracy is still close to 0 on T3 and T4. Some examples of predictions of the MemNN for T1-4 are given in Appendix B. On the OOV tasks again performance is improved, but this is all due to better performance on *known* words, as unknown words are simply not used without the match type features. As stated in Appendix C, optimal hyperparameters on several of the tasks involve 3 or 4 hops, indicating that iterative accessing and reasoning over the conversation helps, e.g. on T3 using 1 hop gives 64.8% while 2 hops yields 74.7%. Appendix B displays illustrative examples of Memory Networks predictions on T 1-4 and Concierge.

Memory Networks *with match type features* give two performance gains over the same models without match type features: (i) T4 (providing information) becomes solvable because matches can be made to the results of the API call; and (ii) out-of-vocabulary results are significantly improved as well. Still, tasks T3 and T5 are still fail cases, performance drops slightly on T2 compared to not using match type features, and no relative improvement is observed on T6. Finally, note that matching words on its own is not enough, as evidenced by the poor performance of TF-IDF matching; this idea must be combined with types and the other properties of the MemNN model.

Unsurprisingly, perfectly coded rule-based systems can solve the simulated tasks T1-T5 perfectly, whereas our machine learning methods cannot. However, it is not easy to build an effective rule-based

system when dealing with real language on real problems, and our rule based system is outperformed by MemNNs on the more realistic task T6.

Overall, while the methods we tried made some inroads into these tasks, there are still many challenges left unsolved. Our best models can learn to track implicit dialog states and manipulate OOV words and symbols (T1-T2) to issue API calls and progress in conversations, but they are still unable to perfectly handle interpreting knowledge about entities (from returned API calls) to present results to the user, e.g. displaying options in T3. The improvement observed on the simulated tasks e.g. where MemNNs outperform supervised embeddings which in turn outperform IR methods, is also seen on the realistic data of T6 with similar relative gains. This is encouraging as it indicates that future work on breaking down, analysing and developing models over the simulated tasks should help in the real tasks as well. Results on Concierge confirm this observation: the pattern of relative performances of methods is the same on Concierge and on our series of tasks. This suggests that our synthetic data can indeed be used as an effective evaluation proxy.

## 6  CONCLUSION

We have introduced an open dataset and task set for evaluating end-to-end goal-oriented dialog learning methods in a systematic and controlled way. We hope this will help foster progress of end-to-end conversational agents because (i) existing measures of performance either prevent reproducibility (different Mechanical Turk jobs) or do not correlate well with human judgements (Liu *et al.*, 2016); (ii) the breakdown in tasks will help focus research and development to improve the learning methods; and (iii) goal-oriented dialog has clear utility in real applications. We illustrated how to use the testbed using a variant of end-to-end Memory Networks, which prove an effective model on these tasks relative to other baselines, but are still lacking in some key areas.

ACKNOWLEDGMENTS

The authors would like to thank Martin Raison, Alex Lebrun and Laurent Landowski for their help with the Concierge  data.

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

## A    MEMORY NETWORKS IMPLEMENTATION

**Storing and representing the conversation history**    As the model conducts a conversation with the user, at each time step $t$ the previous utterance (from the user) and response (from the model) are appended to the memory. Hence, at any given time there are $c_1^u, \dots c_t^u$ user utterances and $c_1^r, \dots c_{t-1}^r$ model responses stored (i.e. the entire conversation).[2] The aim at time $t$ is to thus choose the next response $c_t^r$. We train on existing full dialog transcripts, so at training time we know the upcoming utterance $c_t^r$ and can use it as a training target. Following Dodge *et al.* (2016), we represent each utterance as a bag-of-words and in memory it is represented as a vector using the embedding matrix $A$, i.e. the memory is an array with entries:

$$m = (A\Phi(c_1^u), A\Phi(c_1^r) \dots, A\Phi(c_{t-1}^u), A\Phi(c_{t-1}^r))$$

where $\Phi(\cdot)$ maps the utterance to a bag of dimension $V$ (the vocabulary), and $A$ is a $d \times V$ matrix, where $d$ is the embedding dimension. We retain the last user utterance $c_t^u$ as the "input" to be used directly in the controller. The contents of each memory slot $m_i$ so far does not contain any information of which speaker spoke an utterance, and at what time during the conversation. We therefore encode both of those pieces of information in the mapping $\Phi$ by extending the vocabulary to contain $T = 1000$ extra "time features" which encode the index $i$ into the bag-of-words, and two more features that encode whether the utterance was spoken by the user or the model.

**Attention over the memory**    The last user utterance $c_t^u$ is embedded using the same matrix $A$ giving $q = A\Phi(c_t^u)$, which can also be seen as the initial state of the controller. At this point the controller reads from the memory to find salient parts of the previous conversation that are relevant to producing a response. The match between $q$ and the memories is computed by taking the inner product followed by a softmax: $p_i = \text{Softmax}(u^\top m_i)$, giving a probability vector over the memories. The vector that is returned back to the controller is then computed by $o = R \sum_i p_i m_i$ where $R$ is a $d \times d$ square matrix. The controller state is then updated with $q_2 = o + q$. The memory can be iteratively reread to look for additional pertinent information using the updated state of the controller $q_2$ instead of $q$, and in general using $q_h$ on iteration $h$, with a fixed number of iterations $N$ (termed $N$ hops). Empirically we find improved performance on our tasks with up to 3 or 4 hops.

**Choosing the response**    The final prediction is then defined as:

$$\hat{a} = \text{Softmax}(q_{N+1}^\top W\Phi(y_1), \dots, q_{N+1}^\top W\Phi(y_C))$$

where there are $C$ candidate responses in $y$, and $W$ is of dimension $d \times V$. In our tasks the set $y$ is a (large) set of candidate responses which includes all possible bot utterances and API calls.

The entire model is trained using stochastic gradient descent (SGD), minimizing a standard cross-entropy loss between $\hat{a}$ and the true label $a$.

## B    EXAMPLES OF PREDICTIONS OF A MEMORY NETWORK

Tables 3, 4, 5 and 6 display examples of predictions of the best performing Memory Network on full dialogs, Task 5, (with 3 hops) on test examples of Tasks 1-4 along with the values of the attention over each memory for each hop ($p_i$ as defined in Sec. A). This model does not use match type features. Then, Table 7 displays an example of prediction of the best performing Memory Network on Concierge (with 2 hops) on a test example along with the values of the attention over each memory for each hop.

## C    HYPERPARAMETERS

Tables 8 and 9 respectively display the values of the hyperparameters of the best Supervised Embeddings and Memory Networks selected for each task. These models were selected using the best validation validation sets.

## D    ADDITIONAL RESULTS

Table 10 provides results for additional variants of supervised embeddings, using either a dictionary that includes all bigrams to leverage some word order information, or match type features. On some tasks, supervised embeddings perform better when the last user utterance is used as sole input, without the full dialog history (see Table 8). When no history is used, we slightly adapt match type features to only record type: a special word corresponding to type $T$ (e.g., phone, address, etc) is appended to the representation of a candidate if the

---

[2]API calls are stored as bot utterances $c_i^r$, and KB facts resulting from such calls as user utterances $c_i^u$.

Table 3: **Task 1 (Issue API call)** The model learns to direct its attention towards the 4 memories containing the information key to issue the API call. More hops help to strengthen this signal. <silence> is a special token used to indicate that the user did not speak at this turn – the model has to carry out the conversation with no additional input.

| Time | Locutor | Dialog History | Hop #1 | Hop #2 | Hop #3 |
|------|---------|----------------|--------|--------|--------|
| 1 | User | hi | .054 | .016 | .020 |
| 2 | Bot | hello what can i help you with today | .040 | .024 | .008 |
| 3 | User | may i have a table in paris | .099 | .137 | .141 |
| 4 | Bot | i'm on it | .048 | .028 | .004 |
| 5 | User | <silence> | .181 | .099 | .037 |
| 6 | Bot | any preference on a type of cuisine | .056 | .090 | .014 |
| 7 | User | i love indian food | .159 | .188 | .238 |
| 8 | Bot | how many people would be in your party | .051 | .022 | .010 |
| 9 | User | we will be six | .129 | .144 | .114 |
| 10 | Bot | which price range are looking for | .039 | .028 | .006 |
| 11 | User | in a moderate price range please | .128 | .213 | .405 |
| 12 | Bot | ok let me look into some options for you | .016 | .011 | .003 |
| **User input** | | <silence> | | | |
| **Correct answer** | | `api_call indian paris six moderate` | | | |
| **Predicted answer** | | `api_call indian paris six moderate` | **[Correct]** | | |

Table 4: **Task 2 (Update API call)** Out of the multiple memories from the current dialog, the model correctly focuses on the 2 important pieces: the original API call and the utterance giving the update.

| Time | Locutor | Dialog history | Hop #1 | Hop #2 | Hop #3 |
|------|---------|----------------|--------|--------|--------|
| 3 | User | may i have a table in paris | .061 | .072 | .040 |
| 4 | Bot | i'm on it | .026 | .012 | .001 |
| 5 | User | <silence> | .087 | .042 | .012 |
| 6 | Bot | any preference on a type of cuisine | .026 | .023 | .001 |
| 7 | User | i love indian food | .081 | .070 | .055 |
| 8 | Bot | how many people would be in your party | .025 | .006 | .001 |
| 9 | User | we will be six | .059 | .051 | .018 |
| 10 | Bot | which price range are looking for | .038 | .043 | .004 |
| 11 | User | in a moderate price range please | .080 | .095 | .096 |
| 12 | Bot | ok let me look into some options for you | .025 | .042 | .003 |
| 13 | User | <silence> | .127 | .069 | .032 |
| 14 | Bot | `api_call indian paris six moderate` | .062 | .113 | .043 |
| 15 | User | instead could it be with french food | .188 | .311 | .683 |
| 16 | Bot | sure is there anything else to update | .016 | .007 | .001 |
| 17 | User | no | .028 | .013 | .007 |
| 18 | Bot | ok let me look into some options for you | .011 | .006 | .000 |
| **User input** | | <silence> | | | |
| **Correct answer** | | `api_call french paris six moderate` | | | |
| **Predicted answer** | | `api_call french paris six moderate` | **[Correct]** | | |

Table 5: **Task 3 (Displaying options)** The model knows it has to display options but the attention is wrong: it should attend on the ratings to select the best option (with highest rating). It cannot learn that properly and match type features do not help. It is correct here by luck, the task is not solved overall (see Tab. 2). We do not show all memories in the table, only those with meaningful attention.

| Time | Locutor | Dialog history | Hop #1 | Hop #2 | Hop #3 |
|---|---|---|---|---|---|
| 14 | Bot | api_call indian paris six moderate | .012 | .000 | .000 |
| 15 | User | instead could it be with french food | .067 | .103 | .147 |
| 20 | Bot | api_call french paris six moderate | .012 | .000 | .000 |
| 21 | User | resto_1 r_phone rest_1_phone | .018 | .004 | .000 |
| 23 | User | resto_1 r_cuisine french | .029 | .005 | .000 |
| 24 | User | resto_1 r_location paris | .060 | .292 | .094 |
| 25 | User | resto_1 r_number six | .050 | .298 | .745 |
| 26 | User | resto_1 r_price moderate | .060 | .090 | .002 |
| 27 | User | resto_1 r_rating 6 | .016 | .002 | .000 |
| 30 | User | resto_2 r_cuisine french | .031 | .007 | .000 |
| 31 | User | resto_2 r_location paris | .040 | .081 | .004 |
| 32 | User | resto_2 r_number six | .020 | .012 | .000 |
| 33 | User | resto_2 r_price moderate | .029 | .009 | .000 |
| 37 | User | resto_3 r_cuisine french | .014 | .001 | .000 |
| 38 | User | resto_3 r_location paris | .028 | .016 | .001 |
| 39 | User | resto_3 r_number six | .024 | .022 | .004 |
| 40 | User | resto_3 r_price moderate | .039 | .015 | .001 |
| **User input** | | <silence> | | | |
| **Correct answer** | | what do you think of this option: resto_1 | | | |
| **Predicted answer** | | what do you think of this option: resto_1 | **[Correct]** | | |

Table 6: **Task 4 (Providing extra-information)** The model knows it must display a phone or an address, but, as explained in Section A the embeddings mix up the information and make it hard to distinguish between different phone numbers or addresses, making answering correctly very hard. As shown in the results of Tab. 2, this problem can be solved by adding match type features, that allow to emphasize entities actually appearing in the history. The attention is globally wrong here.

| Time | Locutor | Dialog history | Hop #1 | Hop #2 | Hop #3 |
|---|---|---|---|---|---|
| 14 | Bot | api_call indian paris six moderate | .006 | .000 | .000 |
| 15 | User | instead could it be with french food | .024 | .011 | .007 |
| 20 | Bot | api_call french paris six moderate | .005 | .000 | .001 |
| 21 | User | resto_1 r_phone resto_1_phone | .011 | .005 | .004 |
| 22 | User | resto_1 r_address resto_1_address | .018 | .004 | .001 |
| 23 | User | resto_1 r_cuisine french | .018 | .003 | .001 |
| 24 | User | resto_1 r_location paris | .068 | .091 | .108 |
| 25 | User | resto_1 r_number six | .086 | .078 | .020 |
| 26 | User | resto_1 r_price moderate | .070 | .225 | .369 |
| 27 | User | resto_1 r_rating 6 | .014 | .006 | .008 |
| 28 | User | resto_2 r_phone resto_2_phone | .015 | .009 | .006 |
| 29 | User | resto_2 r_address resto_2_address | .014 | .004 | .001 |
| 31 | User | resto_2 r_location paris | .075 | .176 | .193 |
| 32 | User | resto_2 r_number six | .100 | .126 | .026 |
| 33 | User | resto_2 r_price moderate | .038 | .090 | .167 |
| 35 | User | resto_3 r_phone resto_3_phone | .004 | .001 | .001 |
| 36 | User | resto_3 r_address resto_3_address | .005 | .002 | .001 |
| 37 | User | resto_3 r_location paris | .028 | .028 | .026 |
| 39 | User | resto_3 r_number six | .039 | .013 | .002 |
| 40 | User | resto_3 r_price moderate | .018 | .008 | .013 |
| 42 | Bot | what do you think of this option: resto_1 | .074 | .001 | .000 |
| 43 | User | let's do it | .032 | .004 | .001 |
| 44 | Bot | great let me do the reservation | .003 | .000 | .000 |
| **User input** | | do you have its address | | | |
| **Correct answer** | | here it is resto_1_address | | | |
| **Predicted answer** | | here it is: resto_8_address | **[Incorrect]** | | |

Table 7: **Concierge Data** The model is also able to learn from human-human dialogs. <person>, <org>, <number> and <date> are special tokens used to anonymize the data. We report the top 5 answers predicted by the model. They are all semantically equivalent. Note that the utterances, while all produced by humans, are not perfect English ("rservation", "I'll check into it")

| Time | Locutor | Dialog History | Hop #1 | Hop #2 |
|------|---------|---------------|--------|--------|
| 1 | User | hey concierge | .189 | .095 |
| 2 | User | could you check if i can get a rservation at <org> <date> for brunch | .209 | .178 |
| 3 | User | <number> people | .197 | .142 |
| 4 | User | <silence> | .187 | .167 |
| 5 | Bot | hi <person> unfortunately <org> is fully booked for <date> and there's <number> people on the waiting list | .225 | .410 |
| **User input** | | when's the earliest availability | | |
| **Correct answer** | | i'll check | | |
| **Pred. answer #1** | | i'm on it | [Incorrect] | |
| **Pred. answer #2** | | i'll find out | [Incorrect] | |
| **Pred. answer #3** | | i'll take a look | [Incorrect] | |
| **Pred. answer #4** | | i'll check | **[Correct]** | |
| **Pred. answer #5** | | i'll check into it | [Incorrect] | |

Table 8: **Hyperparameters of Supervised Embeddings.** When Use History is True, the whole conversation history is concatenated with the latest user utterance to create the input. If False, only the latest utterance is used as input.

| Task | Learning Rate | Margin $m$ | Embedding Dim $d$ | Negative Cand. $N$ | Use History |
|------|--------------|-----------|-------------------|--------------------|-------------|
| Task 1 | 0.01 | 0.01 | 32 | 100 | True |
| Task 2 | 0.01 | 0.01 | 128 | 100 | False |
| Task 3 | 0.01 | 0.1 | 128 | 1000 | False |
| Task 4 | 0.001 | 0.1 | 128 | 1000 | False |
| Task 5 | 0.01 | 0.01 | 32 | 100 | True |
| Task 6 | 0.001 | 0.01 | 128 | 100 | False |
| Concierge | 0.001 | 0.1 | 64 | 100 | False |

Table 9: **Hyperparameters of Memory Networks.** The longer and more complex the dialogs are, the more hops are needed.

| Task | Learning Rate | Margin $m$ | Embedding Dim $d$ | Negative Cand. $N$ | Nb Hops |
|------|--------------|-----------|-------------------|--------------------|---------|
| Task 1 | 0.01 | 0.1 | 128 | 100 | 1 |
| Task 2 | 0.01 | 0.1 | 32 | 100 | 1 |
| Task 3 | 0.01 | 0.1 | 32 | 100 | 3 |
| Task 4 | 0.01 | 0.1 | 128 | 100 | 2 |
| Task 5 | 0.01 | 0.1 | 32 | 100 | 3 |
| Task 6 | 0.01 | 0.1 | 128 | 100 | 4 |
| Concierge | 0.001 | 0.1 | 128 | 100 | 2 |

candidate contains a word that appears in the knowledge base as an entity of type $T$, regardless of whether the same word appeared earlier in the conversation. As seen on Table 10, match type features improve performance on out-of-vocabulary tasks 1 and 5, bringing it closer to that of Memory Networks without match type features, but still quite lagging Memory Networks with match type features. Bigrams slightly hurt rather than help performance, except in Task 5 in the standard in-vocabulary setup (performance is lower in the OOV setup).

Table 10: **Test results across all tasks and methods.** For tasks T1-T5 results are given in the standard setup and the out-of-vocabulary (OOV) setup, where words (e.g. restaurant names) may not have been seen during training. Task T6 is the Dialog state tracking 2 task with real dialogs, and only has one setup. Best performing methods (or methods within 0.1% of best performing) are given in bold for the per-response accuracy metric, with the per-dialog accuracy given in parenthesis.

| Task | Supervised Embeddings | | | | | | Memory Networks | | | |
|---|---|---|---|---|---|---|---|---|---|---|
| | no match type no bigram | | + match type no bigram | | + bigrams no match type | | no match type | | + match type | |
| T1: Issuing API calls | **100** | (100) | 83.2 | (0) | 98.6 | (92.4) | **99.9** | (99.6) | **100** | (100) |
| T2: Updating API calls | 68.4 | (0) | 68.4 | (0) | 68.3 | (0) | **100** | (100) | 98.3 | (83.9) |
| T3: Displaying options | 64.9 | (0) | 64.9 | (0) | 64.9 | (0) | **74.9** | (2.0) | **74.9** | (0) |
| T4: Providing information | 57.2 | (0) | 57.2 | (0) | 57.3 | (0) | 59.5 | (3.0) | **100** | (100) |
| T5: Full dialogs | 75.4 | (0) | 76.2 | (0) | 83.4 | (0) | **96.1** | (49.4) | 93.4 | (19.7) |
| T1(OOV): Issuing API calls | 60.0 | (0) | 67.2 | (0) | 58.8 | (0) | 72.3 | (0) | **96.5** | (82.7) |
| T2(OOV): Updating API calls | 68.3 | (0) | 68.3 | (0) | 68.3 | (0) | 78.9 | (0) | **94.5** | (48.4) |
| T3(OOV): Displaying options | 65.0 | (0) | 65.0 | (0) | 62.1 | (0) | 74.4 | (0) | **75.2** | (0) |
| T4(OOV): Providing inform. | 57.0 | (0) | 57.1 | (0) | 57.0 | (0) | 57.6 | (0) | **100** | (100) |
| T5(OOV): Full dialogs | 58.2 | (0) | 64.4 | (0) | 50.4 | (0) | 65.5 | (0) | **77.7** | (0) |
| T6: Dialog state tracking 2 | 22.6 | (0) | 22.1 | (0) | 21.8 | (0) | **41.1** | (0) | **41.0** | (0) |

