# Peer review of "Learning End-to-End Goal-Oriented Dialog"

_ICLR 2017 — accepted_

[Official Review · AnonReviewer3 · rating 8 · confidence 5 · 16 Dec 2016 (modified: 21 Jan 2017)]

This paper presents a new, public dataset and tasks for goal-oriented dialogue applications. The dataset and tasks are constructed artificially using rule-based programs, in such a way that different aspects of dialogue system performance can be evaluated ranging from issuing API calls to displaying options, as well as full-fledged dialogue.

This is a welcome contribution to the dialogue literature, which will help facilitate future research into developing and understanding dialogue systems. Still, there are pitfalls in taking this approach. First, it is not clear how suitable Deep Learning models are for these tasks compared to traditional methods (rule-based systems or shallow models), since Deep Learning models are known to require many training examples and therefore performance difference between different neural networks may simply boil down to regularization techniques. The tasks 1-5 are also completely deterministic, which means evaluating performance on these tasks won't measure the ability of the models to handle noisy and ambiguous interactions (e.g. inferring a distribution over user goals, or executing dialogue repair strategies), which is a very important aspect in dialogue applications. Overall, I still believe this is an interesting direction to explore.

As discussed in the comments below, the paper does not have any baseline model with word order information. I think this is a strong weakness of the paper, because it makes the neural networks appear unreasonably strong, yet simpler baselines could very likely be be competitive (or better) than the proposed neural networks. To maintain a fair evaluation and correctly assess the power of representation learning for this task, I think it's important that the authors experiment with one additional non-neural network benchmark model which takes into account word order information. This would more convincly demonstrate the utility of Deep Learning models for this task. For example, the one could experiment with a logistic regression model which takes as input 1) word embeddings (similar to the Supervised Embeddings model), 2) bi-gram features, and 3) match-type features. If such a baseline is included, I will increase my rating to 8.



Final minor comment: in the conclusion, the paper states "the existing work has no well defined measures of performances". This is not really true. End-to-end trainable models for task-oriented dialogue have well-defined performance measures. See, for example "A Network-based End-to-End Trainable Task-oriented Dialogue System" by Wen et al. On the other hand, non-goal-oriented dialogue are generally harder to evaluate, but given human subjects these can also be evaluated. In fact, this is what Liu et al (2016) do for Twitter. See also, "Strategy and Policy Learning for Non-Task-Oriented Conversational Systems" by Yu et al.

----

I've updated my score following the new results added in the paper.

[Official Review · AnonReviewer4 · rating 7 · confidence 4 · 17 Dec 2016 (modified: 24 Jan 2017)]

SYNOPSIS:
This paper introduces a new dataset for evaluating end-to-end goal-oriented dialog systems.  All data is generated in the restaurant setting, where the goal is to find availability and eventually book a table based on parameters provided by the user to the bot as part of a dialog.  Data is generated by running a simulation using an underlying knowledge base to generate samples for the different parameters (cuisine, price range, etc), and then applying rule-based transformations to render natural language descriptions. The objective is to rank a set of candidate responses for each next turn of the dialog, and evaluation is reported in terms of per-response accuracy and per-dialog accuracy. The authors show that Memory Networks are able to improve over basic bag-of-words baselines.

THOUGHTS:
I want to thank the authors for an interesting contribution.  Having said that, I am skeptical about the utility of end-to-end trained systems in the narrow-domain setting. In the open-domain setting, there is a strong argument to be made that hand-coding all states and responses would not scale, and hence end-to-end trained methods make a lot of sense. However, in the narrow-domain setting, we usually know and understand the domain quite well, and the goal is to obtain high user satisfaction. Doesn't it then make sense in these cases to use the domain knowledge to engineer the best system possible?

Given that the domain is already restricted, I'm also a bit disappointed that the goal is to RANK instead of GENERATE responses, although I understand that this makes evaluation much easier. I'm also unsure how these candidate responses would actually be obtained in practice? It seems that the models rank the set of all responses in train/val/test (last sentence before Sec 3.2). Since a key argument for the end-to-end training approach is ease of scaling to new domains without having to manually re-engineer the system, where is this information obtained for a new domain in practice?  Generating responses would allow much better generalization to new domains, as opposed to simply ranking some list of hand-collected generic responses, and in my mind this is the weakest part of this work.

Finally, as data is generated using a simulation by expanding (cuisine, price, ...) tuples using NL-generation rules, it necessarily constrains the variability in the training responses. Of course, this is traded off with the ability to generate unlimited data using the simulator. But I was unable to see the list of rules that was used. It would be good to publish this as well.

Overall, despite my skepticism, I think it is an interesting contribution worthy of publication at the conference. 

------

I've updated my score following the clarifications and new results.

[Official Review · AnonReviewer2 · rating 8 · confidence 4 · 18 Dec 2016]
**Thought provoking paper, more on the metrics than the algorithms.**

Attempts to use chatbots for every form of human-computer interaction has been a major trend in 2016, with claims that they could solve many forms of dialogs beyond simple chit-chat. This paper represents a serious reality check. While it is mostly relevant for Dialog/Natural Language venues (to educate software engineer about the limitations of current chatbots), it can also be published at Machine Learning venues (to educate researchers about the need for more realistic validation of ML applied to dialogs), so I would consider this work of  high significance.

Two important conjectures are underlying this paper and likely to open to more research. While they are not in writing, Antoine Bordes clearly stated them during a NIPS workshop presentation that covered this work. Considering the metrics chosen in this paper:
1)	The performance of end2end ML approaches is still insufficient for goal oriented dialogs.
2)	When comparing algorithms, relative performance on synthetic data is a good predictor of performance on natural data. This would be quite a departure from previous observations, but the authors made a strong effort to match the synthetic and natural conditions.

While its original algorithmic contribution consists in one rather simple addition to memory networks (match type), it is the first time these are deployed and tested on a goal-oriented dialog, and the experimental protocol is excellent. The overall paper clarity is excellent and accessible to a readership beyond ML and dialog researchers. I was in particular impressed by how the short appendix on memory networks summarized them so well, followed by the tables that explained the influence of the number of hops.

While this paper represents the state-of-the-art in the exploration of more rigorous metrics for dialog modeling, it also reminds us how brittle and somewhat arbitrary these remain. Note this is more a recommendation for future research than  for revision.

First they use the per-response accuracy (basically the next utterance classification among a fixed list of responses). Looking at table 3 clearly shows how absurd this can be in practice: all that matters is a correct API call and a reasonably short dialog, though this would only give us a 1/7 accuracy, as the 6 bot responses needed to reach the API call also have to be exact.

Would the per-dialog accuracy, where all responses must be correct, be better? Table 2 shows how sensitive it is to the experimental protocol. I was initially puzzled that the accuracy for subtask T3 (0.0) was much lower that the accuracy for the full dialog T5 (19.7), until the authors pointed me to the tasks definitions (3.1.1) where T3 requires displaying 3 options while T5 only requires displaying one.

For the concierge data, what would happen if ‘correct’ meant being the best, not among the 5-best? 

While I cannot fault the authors for using standard dialog metrics, and coming up with new ones that are actually too pessimistic, I can think of one way to represent dialogs that could result in more meaningful metrics in goal oriented dialogs. Suppose I sell Virtual Assistants as a service, being paid upon successful completion of a dialog. What is the metric that would maximize my revenue? In this restaurant problem, the loss would probably be some weighted sum of the number of errors in the API call, the number of turns to reach that API call and the number of rejected options by the user. However, such as loss cannot be measured on canned dialogs and would either require a real human user or an realistic simulator

Another issue closely related to representation learning that this paper fails to address or explain properly is what happens if the vocabulary used by the user does not match exactly the vocabulary in the knowledge base. In particular, for the match type algorithm to code ‘Indian’ as ‘type of cuisine’, this word would have to occur exactly in the KB. I can imagine situations where the KB uses some obfuscated terminology, and we would like ML to learn the associations rather than humans to hand-describe them.

[Author Response · Antoine Bordes · 14 Jan 2017]
**General response to reviewers**

We thank the reviewers for their thorough and insightful comments on the paper. We address them in individual rebuttals. We also updated the paper to add more experimental results and clarifications as requested.

[Reviewer Comment · AnonReviewer3 · 24 Jan 2017]
**Simpler Models Achieve Better Performance**

This paper was submitted to arXiv last week:

[Final Decision · Program Chairs · 06 Feb 2017]
**ICLR committee final decision**

Most dialog systems are based on chit-chat models. This paper explores goal-directed conversations, such as those that arise in booking a restaurant. While the methodology is rather thin, this is not the main focus of the paper. The authors provide creative evaluation protocols, and datasets. The reviewers liked the paper, and so does the AC. The paper will make a nice oral, on a topic that is largely explored but opening a direction that is quite novel.